# (+)-Catechin Stereoisomer and Gallate Induce Oxidative Stress in Rat Aorta

**DOI:** 10.3390/molecules27113379

**Published:** 2022-05-24

**Authors:** Tamara V. Samokhvalova, Yuri A. Kim, Antonia F. Korystova, Ludmila N. Kublik, Vera V. Shaposhnikova, Yuri N. Korystov

**Affiliations:** 1Institute of Theoretical and Experimental Biophysics, Russian Academy of Sciences, Institutskaya 3, 142290 Pushchino, Russia; tomikasd@mail.ru (T.V.S.); maneris@rambler.ru (A.F.K.); ludkub@rambler.ru (L.N.K.); vera_shaposh@rambler.ru (V.V.S.); 2Institute of Cell Biophysics, Russian Academy of Sciences, Institutskaya 3, 142290 Pushchino, Russia; yuk01@rambler.ru

**Keywords:** aorta, angiotensin-converting enzyme, gallate, stereoisomers of catechin, reactive oxygen species

## Abstract

The goal of the work was to study changes in the activity of the angiotensin-converting enzyme (ACE) and production of reactive oxygen species (ROS) in the aorta of rats after the intraperitoneal injection of stereoisomers of catechin and gallate. The activity of ACE in the aorta sections was determined by measuring the hydrolysis of hippuryl-l-histidyl-l-leucine. The production of ROS in the aorta sections was estimated from the oxidation of dichlorodihydrofluorescein. The time and dose dependences of the effect of catechin stereoisomers and gallate on ACE activity and ROS production in the aorta were studied. It was shown that (+)-catechin and gallate increased the ACE activity and ROS production, and (−)-catechin and (−)-epicatechin did not influence these parameters. The doses of (+)-catechin and gallate that increased the ACE activity to a half-maximal value (AD_50_) were 0.04 and 0.03 µg/kg, respectively. Fucoidin, a blocker of leukocyte adhesion to the endothelium, reduced the ACE activity to the control level in the aortas of (+)-catechin-treated rats.

## 1. Introduction

Atherosclerosis is, at present, the main cause of cardiovascular diseases (CVD) associated with mortality [1]. It is initiated by oxidative stress in the vessels [2]. Oxidative stress is caused by the activation of the angiotensin-converting enzyme (ACE) and an increase in the concentration of its product, angiotensin II [3,4]. Angiotensin II activates NADPH oxidase [5,6], whose stimulation enhances the production of reactive oxygen species (ROS), which, in turn, activates atherosclerosis progression [7,8,9].

Population studies have shown that three cups of green or black tea a day significantly reduce the risk of CVDs [10]. Green tea contains about 30% of the catechins of the dry weight of a leaf, and 75% of the catechins are due to galloylated catechins: (−)-epigallocatechin galate and (−)-epigallocatechin [11]. Catechins possess various biological activities: antioxidative, antimicrobial, anti-allergic, antidiabetic, anti-inflammatory, anticancer, chemoprotective, neuroprotective, and immunomodulatory [12,13]. Epigallocatechin gallate reduces blood pressure, the level of blood cholesterol, the amount of body fat, and the risk of osteoporotic fractures [13,14].

We have shown earlier that tea prevents radiation-induced oxidative stress in the aortas of rats [15]. Black tea was found to be more effective than green tea, which was explained by a low content of galloylated catechins in black tea, because galloylated catechins induce oxidative stress [15]. Green tea contains (+)-catechin and galloylated (−)-epicatechin [16]. It was also shown that a mixture of (+)- and (−)-catechins increases ACE activity in the aortas of rats [17]. In the present work, to determine which components of green tea-galloylated catechins are responsible for oxidative stress in the rat aorta, we studied the effects of (+)-, (−)-catechins, (−)-epicatechin, and gallate (Figure 1) on ACE activity and ROS production.

## 2. Results

### 2.1. Dynamics of Changes in the Activity of ACE after Injection of (+)-Catechin

In order to determine the optimal time of treatment of rats with flavonoids, the dynamics of the changes in ACE activity were studied. Figure 2 shows that ACE activity increases to a maximum by 3 h after an injection of 1 µg/kg of (+)-catechin; after which, it decreases to the control level during the day.

### 2.2. Dose Dependences of the Effect of Catechins and Gallate on ACE Activity

Figure 3 and Figure 4 show the dose dependences of the effects of catechin stereoisomers (Figure 3) and gallate (Figure 4). It is seen from Figure 3 that (−)-catechin and (−)-epicatechin do not change the ACE activity, and (+)-catechin increases it. The dose of (+)-catechin that induces a half-maximum increase in ACE activity in the aorta (AD_50_) is 0.04 μg/kg. Gallate also increases the ACE activity (Figure 4). The AD_50_ for gallate is 0.03 μg/kg.

### 2.3. Effect of (+)-Catechin and Gallate on ROS Production in the Aorta

The influence of (+)-catechin and gallate on ROS production in the aorta was studied at doses that induce the maximum activation of ACE. The data in Figure 5 indicate that both compounds at a dose of 1 µg/kg increase ROS production in the aorta up to 44%, and (+)-catechin at a dose of 3 µg/kg increases it up to 83%.

### 2.4. Effect of Fucoidin on the Increase in ACE Activity in Rat Aortas by the Actions of (+)-Catechin and Gallate

Figure 6 presents the data on the influence of fucoidin, a blocker of leukocyte adhesion to the endothelium, on the increase in ACE activity in the aorta induced by (+)-catechin and gallate. It is seen that fucoidin suppresses the (+)-catechin-induced increase in ACE activity, but it does not influence the increase in ACE activity induced by gallate.

## 3. Discussion

It is well-known that stereoisomers produce different effects in biochemical reactions and living systems due to the selective binding to enzymes, receptors, and DNA [18]. Therefore, they can have different pharmacokinetic, pharmacodynamic, therapeutic, and adverse effects [18]. There is evidence indicating that this also holds true for catechin stereoisomers. A stereochemical configuration significantly affects the transport and metabolism of catechins in cell monolayers [19] and bioavailability in rats [20]. Moreover, catechin stereoisomers can induce even opposite effects. Thus, the addition of 0.5 mM catechin to hepatocytes from fasted rats was shown to result in a 90% stimulation of the net glycogen production in the case of the (+)-isomer and in a 90% inhibition in the case of the (−)-isomer [21]. Our data added information about the different activities of catechin stereoisomers. Catechin stereoisomers bind with different amino acids in enzymes [22], and this fact can explain different influences of these compounds in our work: (+)-catechin activates some processes, inducing oxidative stress, whereas other stereoisomers produce no effect. 

It was shown in the present work and in [15] that (+)-catechin, gallate, and galloylated catechins induce oxidative stress in the aorta. The increase in ACE activity by these compounds is in accordance with the data indicating that catechin and galloylated catechins induce the contraction of rat aortas in vitro [23], because this increase leads to a rise in the concentration of the ACE product angiotensin II vasoconstrictor.

Gallate, (+)-catechin, and galloylated catechins are contained in many natural products: cocoa, tea, blueberries, walnuts, grapes, and a variety of other plants sources [24,25]. The amount of these compounds consumed by people with these products is much greater than the IC_50_ required for the activation of oxidative stress. However, oxidative stress is not initiated, because the same plant sources contain large amounts of flavonoids that suppress oxidative stress: flavanonols, flavonols, and flavones [26]. Dihydroquercetin (flavanonol) cancels oxidative stress in the aorta induced with catechin [17]. The opposite effects on oxidative stress of catechins and flavonols have been studied for tea. Green and black teas contain comparable amounts of flavonols, but green tea contains greater amounts of catechins, about 3.5 times that in black tea [16]. Both teas prevent radiation-induced oxidative stress in the aorta, but black tea is 12 times more effective than green tea [15].

It is possible that green tea diminishes the risk of CVD [10] to a greater extent than the suppression of oxidative stress due to the fact that galloylated catechins decrease other processes involved in atherosclerosis progression. The adhesion of leukocytes and T-lymphocytes to the vascular epithelium [27,28], followed by the activation of endothelial ACE [8], initiates atherosclerosis. The expression of the adhesion molecules ICAM-1 and VCAM-1 in endothelial cells and the adhesion of U937 monocyte cells to them decreases by the action of (−)-epigallocatechin gallate [29,30]. However, very high doses of catechins were used in [29,30] in comparison with the concentration in the blood plasma after the consumption of a cup of tea [11] or a very long treatment [31]. This catechin also induces some effects in animals: it decreases blood pressure in spontaneously hypertensive rats [32,33] and promotes atherosclerotic plaque stability in apolipoprotein E-deficient mice [34]. The doses of (−)-epigallocatechin gallate used in these works were also higher (up to 300 mg/kg). Gallate, as well as (−)-epigallocatechin gallate, also suppress some processes responsible for atherosclerosis progression in spontaneously hypertensive rats [35,36] at large doses. It is possible that the effects of (−)-epigallocatechin gallate are caused by gallate. Thus, it remains unclear whether the effects of (−)-epigallocatechin gallate that suppress atherosclerosis can be accomplished in the human organism upon green tea consumption.

The (+)-catechin-induced increase in ACE activity in the aorta is caused by the adhesion of monocytes to the endothelium, because the adhesion blocker fucoidin suppresses this increase (Figure 6). The effect of gallate on ACE activity is not blocked by fucoidin (Figure 6). This means that the induction of oxidative stress in the aorta with gallate is brought about through another mechanism, which remains unknown. 

## 4. Materials and Methods

### 4.1. Animals, Mode of Introduction of Catechins and Gallate, and Aorta Preparation

Male Wistar rats weighing 300–320 g at an age of 10–11 weeks (*N* = 105) from the animal collection at the Institute of Theoretical and Experimental Biophysics (Pushchino, Russia) were used. The rats were maintained in animal facilities and fed a standard diet with free access to water. All experiments on animals were conducted under protocols approved by the Institute of Theoretical and Experimental Biophysics, Russian Academy of Sciences (protocol number 22/2022 of 5.3.2022). Matrix solutions (1 mg/mL) of all catechins (Sigma, St. Luis, MO, USA) were prepared in a mixture of dimethyl sulfoxide (DMSO) (Sigma, St. Luis, MO, USA) with ethanol at the ratio 1:4, and a matrix solution of gallate Na (Sigma, St. Luis, MO, USA) was prepared in water. The solutions were diluted (in accordance with the dose used) with sterile physiological solution and injected intraperitoneally (volume: 0.6 mL). Rats injected i.p. with physiological saline (0.6 mL) or a diluted mixture of DMSO and ethanol served as a control. The maximal dose of catechins used in the study was 3 µg/kg. At this dose, DMSO was diluted 3000 times and ethanol 750 times. The injection of the physiological solution or a diluted mixture of DMSO with ethanol did not influence the ACE activity in the aortas: control rats, ACE = 30.1 ± 0.8 picomol/min/mm^2^ (*N* = 5); injection of a physiological solution, ACE = 30.1 ± 1.6 picomol/min/mm^2^ (*N* = 3); and a diluted mixture of DMSO and ethanol, ACE = 30.1 ± 1.6 picomol/min/mm^2^ (*N* = 3). Fucoidin (Sigma, St. Luis, MO, USA) (10 μg/kg) was injected intravenously 5 min before the i.p. injection of (+)-catechin or gallate. The aorta sections were prepared as described in [37]. 

### 4.2. Measurements of ACE Activity in the Aorta

The ACE activity was determined by measuring the hydrolysis of hippuryl-l-histidyl-l-leucine (Hip-His-Leu) (Sigma, St. Luis, MO, USA) using the method of Ackermann et al. [38] with a modification by Myamoto et al. [39]. Adetailed description of this method is in Korystova et al. [15]. 

### 4.3. Measurement of ROS in the Aorta

The amount of ROS was determined by the method of Korystov et al. [38].

### 4.4. Statistical Analysis

The results were expressed as the means ± S.E.M. Each experimental point in the figures wasthe result of experiments on three to six animals. The significance of differences in multiple comparisons was determined using the ANOVA and post-hoc Tukey’s tests. *p*-values less than 0.05 were considered significant.

## 5. Conclusions

Based on the results of the study, we conclude that (+)-catechin and gallate increase the activity of ACE and ROS production in the aortas of rats. The other catechin isomers (−)-catechin and (−)-epicatechin do not induce oxidative stress in the aorta. These data point out that oxidative stress induced in the aorta by a mixture of (+)- and (−)-catechins [17] is caused by (+)-catechin, while oxidative stress induced in the aorta by (−)-epigallocatechin and (−)-epigallocatechin gallate [15] arises from gallate.

## Figures and Tables

**Figure 1 molecules-27-03379-f001:**
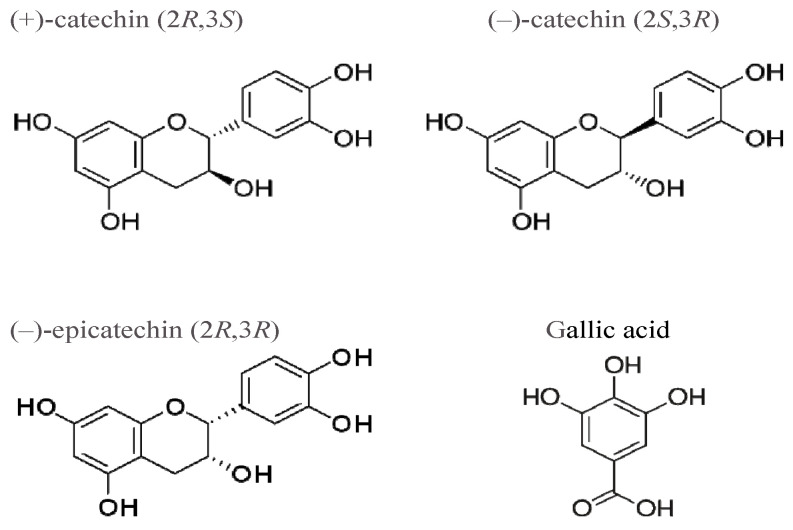
Structural formulas of the compounds used in the study.

**Figure 2 molecules-27-03379-f002:**
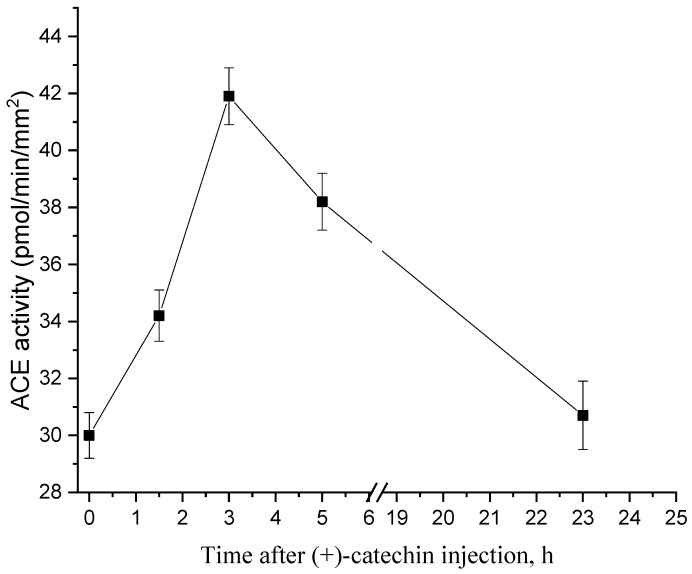
Dynamics of ACE activity in the aorta after the administration of 1 μg/kg of (+)-catechin.

**Figure 3 molecules-27-03379-f003:**
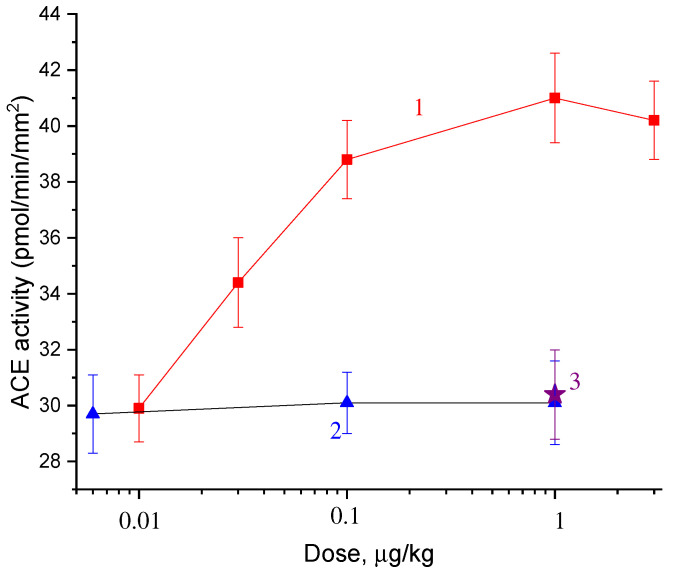
Dose dependences of the effects of (+)-catechin (1), (−)-epicatechin (2), and (−)-catechin on ACE activity in rat aortas.

**Figure 4 molecules-27-03379-f004:**
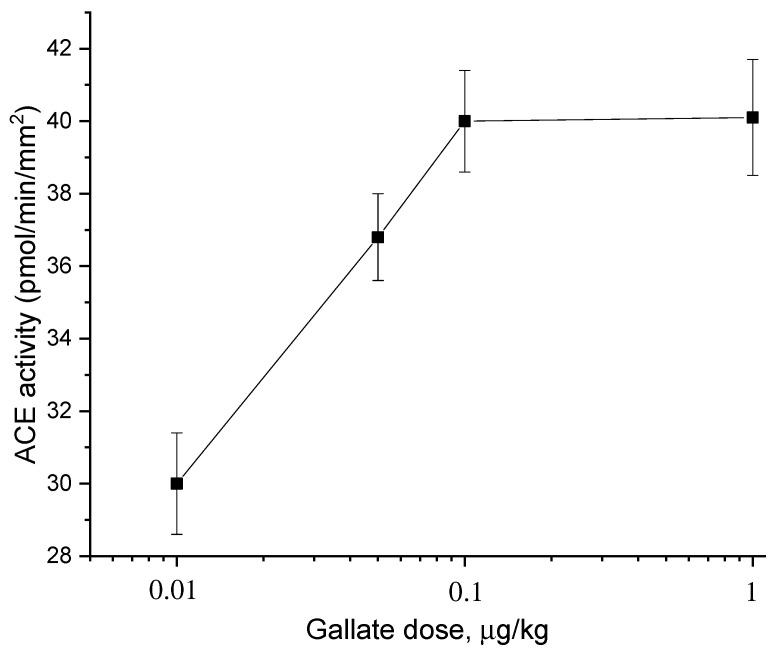
Dose dependences of the effect of gallate on ACE activity in rat aortas.

**Figure 5 molecules-27-03379-f005:**
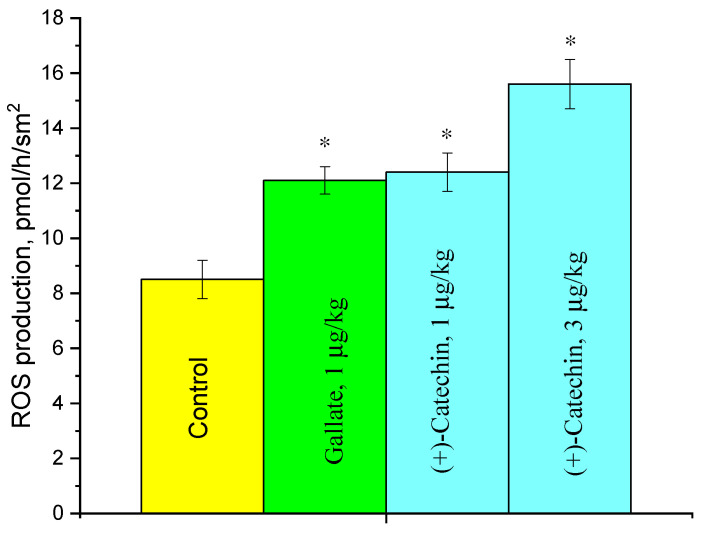
Effects of (+)-catechin and gallate on ROS formation in a rat aorta 3 h after the i.p. injection at a dose of 3 μg/kg. * *p*< 0.05 vs. ROS production in the aortas of control rats.

**Figure 6 molecules-27-03379-f006:**
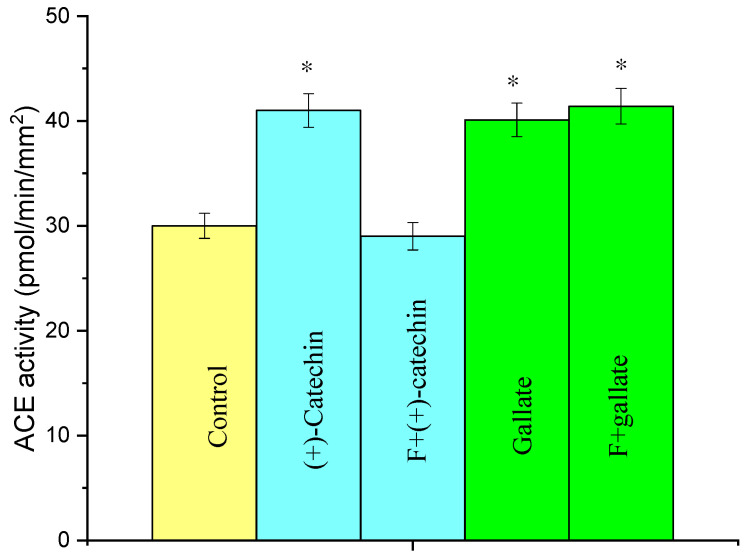
Effect of fucoidin (F) on the increase in ACE activity in the rat aorta by the action of 1 µg/kg of (+)-catechin and gallate. The ACE activity was determined 3 h after the injection of (+)-catechin and gallate. * *p* < 0.05 vs. ACE activity in the aortas of the control rats.

## Data Availability

Not applicable.

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
