# Peer review of "(+)-Catechin Stereoisomer and Gallate Induce Oxidative Stress in Rat Aorta"

_molecules, 2022, doi:10.3390/molecules27113379_

Round 1
Reviewer 1 Report
Authors have reported the (+)-Catechin Stereoisomer and Gallate Induce Oxidative Stress in Rat Aorta but following should be considered
- Authors need to reduce the similiarity index which is more than 50%.
- In the introduction authors should include the different bilogical activities posses by catechin authors may get benefitted by https://doi.org/10.3390/molecules24010156;https://doi.org/10.1002/bkcs.10108 ; https://doi.org/10.5012/jkcs.2021.65.2.106; etc
- Does dose dependent studies done for other compounds viz (-)- catechin etc
- Authors should mention the compound was synthesized or commercially purchased.
- Authors guidelines should be followed references are also not in the format.
- Authors should give some possibe mechanism for the active catechin
Author Response
- Authors need to reduce the similiarity index which is more than 50%.
Answer
The high similarity index is determined by the fact that materials and methods in this work are the same as in previous works. The section Materials and methods in this manuscript makes up 45% of the whole manuscript text. We believe that there is no sense to change the words for synonyms and the order of sentences in the description of the same methods. The similarity index will be decreased as a consequence of an increase in the volume of the Introduction and Discussion according to the recommendations of reviewers.
- In the introduction authors should include the different bilogical activities posses by catechin authors may get benefitted by https://doi.org/10.3390/molecules24010156;https://doi.org/10.1002/bkcs.10108 ; https://doi.org/10.5012/jkcs.2021.65.2.106; etc.
Answer
We included several reviews for the description of different biological activities of catechins and reference 1 - (-)-Epigallocatechin-3-Gallate Decreases Osteoclastogenesis …; reference 2 is not opened (error), and reference 3 is not related to the effects of natural catechins on biological processes.
The text that was added in the manuscript:
Population studies show that three cups of green or black tea a day significantly reduce the risk of CVDs [10]. Green tea contains about 30% of catechins of the dry weight of a leaf, and 75% of catechins are due to galloylated catechins: (–)-epigallocatechin galate and (–)-epigallocatechin [11]. Catechins possess various biological activities: anti-oxidative, antimicrobial, anti-allergic, anti-diabetic, anti-inflammatory, anti-cancer, chemoprotective, neuroprotective, and immunomodulatory [12, 13]. Epigallocatechin gallate reduces blood pressure, the level of blood cholesterol, the amount of body fat, and the risk of osteoporotic fractures [13, 14].
- Does dose dependent studies done for other compounds viz (-)- catechin etc
Answer
Dose dependent studies were done for (+)-catechin, (-)-epicatechin (the present work, Fig. 3), and EGC, EGCG (see reference 15).
- Authors should mention the compound was synthesized or commercially purchased.
Answer
See Materials and methods 4.1. Animals, mode of introduction of catechins and gallate, and aorta preparation, page 6: Matrix solutions (1 mg/ml) of all catechins (Sigma, USA)… and a matrix solution of gallate Na (Sigma, USA). This means that the compounds were commercially purchased.
- Authors guidelines should be followed references are also not in the format.
Answer
Thank you. The format of references was corrected.
- Authors should give some possibe mechanism for the active catechin.
Answer
A possible mechanism for the effect of catechin was suggested in Discussion (Page 5): “The (+)-catechin-induced increase in ACE activity in the aorta is caused by the adhesion of monocytes to the endothelium because the adhesion blocker fucoidin suppresses this increase.” We have no data for a more detail of this mechanism.
We are grateful to the reviewe for the critical remarks. Taking this helpful criticism into account substantially improved the manuscript.
The authors
Reviewer 2 Report
The manuscript “(+) - Catechin Stereoisomer and Gallate Induces Oxidative Stress in Rat Aorta” by Samokhvalova et al. described changes in ACE activity and ROS production in rat aorta of catechin and gallate stereoisomers administered intraperitoneally. From my personal point of view, the manuscript is interesting and the strength of the work is represented by the specificity of action of (+) - catechin and gallate, which increased the activity of ACE and ROS, compared to (-) - catechin and (-) - epicatechin which on the contrary did not affect these parameters. The weak point of the work is to adequately describe in the discussion the probable molecular mechanism that explains the different behavior of the molecules in the experimental model used. The discussion is too short and should be extended, explaining why those doses of the molecules were used. Furthermore, I would like to ask the authors to extrapolate the experimental data obtained in a possible human study. Is there experimental evidence that the pro-oxidant effect of (+) - catechin and gallate has been demonstrated in humans? However, the work is well structured, the figures understandable, the materials and methods well described and the statistical analysis is accurate.
Author Response
Reviewer 2
The weak point of the work is to adequately describe in the discussion the probable molecular mechanism that explains the different behavior of the molecules in the experimental model used (1). The discussion is too short and should be extended, explaining why those doses of the molecules were used (2). Furthermore, I would like to ask the authors to extrapolate the experimental data obtained in a possible human study. Is there experimental evidence that the pro-oxidant effect of (+) - catechin and gallate has been demonstrated in humans?(3)
Answer
- The sentences that were added in Discussion:
It is well known that stereoisomers produce different effects in biochemical reactions and living systems due to the selective binding to enzymes, receptors, and DNA [18] Therefore, they can have different pharmacokinetic, pharmacodynamic, therapeutic, and adverse effects
Catechin stereoisomers bind with different amino acids in enzymes [22], and this fact can explain different influence of these compounds in our work: (+)-catechin activates some processes inducing oxidative stress, whereas other stereoisomers produce no effect.
It was shown in the present work and in [15] that (+)-catechin, gallate, and galloylated catechins induce oxidative stress in the aorta. The increase in ACE activity by these compounds is in accordance with the data indicating that catechin and galloylated catechins induce the contraction of rat aorta in vitro [23], because this increase leads to a rise in the concentration of the ACE product: angiotensin II vasoconstrictor.
Gallate, (+)-catechin, and galloylated catechins are contained in many natural products: cocoa, tea, blueberries, walnuts, grapes, and a variety of other plants sources [24, 25]. The amount of these compounds consumed by people with these products is much greater than IC50 required for the activation of oxidative stress. However, oxidative stress is not initiated because the same plant sources contain large amounts of flavonoids that suppress oxidative stress: flavanonols, flavonols, flavones [26]. Dihydroquercetin (flavanonol) cancels oxidative stress in the aorta induced with catechin [17]. The opposite effects on oxidative stress of catechins and flavonols have been studied for tea. Green and black teas contain comparable amounts of flavonols, but green tea contains greater amounts of catechins, about 3.5 times that in black tea [16]. Both teas prevent radiation-induced oxidative stress in the aorta, but black tea is 12 times more effective than green tea [15].
- The doses of the compounds used were those at which the effect appeared and reached a maximum.
- The people do not consume (+) - catechin and gallate as pure substances but in composition with other flavonoids with natural products only, and in this case possible induction of oxidative stress with (+) - catechin and gallate is cancelled by other flavonoids, for instance, flavonols (see answer 1, last paragraph).
We are grateful to the reviewer for the critical remarks. Taking this helpful criticism into account substantially improved the manuscript.
The authors
Round 2
Reviewer 1 Report
Similiarity index is still is 56 percent
Author Response
Response to Reviewer 1 Comments
Point 1: Authors need to reduce the similiarity index which is more than 50%.
Response 1: The high similarity index is determined by the fact that materials and methods in this work are the same as in previous works. The section Materials and methods in this manuscript makes up 45% of the whole manuscript text. We cannot describe Materials and methods with other words and using another sequence of manipulations because we used the same animals, the same equipment, same drugs, and the same sequence of manipulations. Therefore, we deleted the detailed description of the methods and for the acquaintance with the methods and gave the references to previous works. The text in Introduction and Discussion marked in yellow was rewritten to reduce similarity. The similarity index also decreased as a consequence of an increased volume of the Introduction and Discussion sections.
Point 2: In the introduction authors should include the different bilogical activities posses by catechin authors may get benefitted by https://doi.org/10.3390/molecules24010156;https://doi.org/10.1002/bkcs.10108 ; https://doi.org/10.5012/jkcs.2021.65.2.106; etc.
Response 2: We included several reviews for the description of different biological activities of catechins and reference 1 - (-)-Epigallocatechin-3-Gallate Decreases Osteoclastogenesis …; reference 2 is not opened (error), and reference 3 is not related to the effects of natural catechins on biological processes. The text that was added in the manuscript: Population studies show that three cups of green or black tea a day significantly reduce the risk of CVDs [10]. Green tea contains about 30% of catechins of the dry weight of a leaf, and 75% of catechins are due to galloylated catechins: (–)-epigallocatechin galate and (–)-epigallocatechin [11]. Catechins possess various biological activities: anti-oxidative, antimicrobial, anti-allergic, anti-diabetic, anti-inflammatory, anti-cancer, chemoprotective, neuroprotective, and immunomodulatory [12, 13]. Epigallocatechin gallate reduces blood pressure, the level of blood cholesterol, the amount of body fat, and the risk of osteoporotic fractures [13, 14].
Point 3: Does dose dependent studies done for other compounds viz (-)- catechin etc
Response 3: Dose dependent studies were done for (+)-catechin, (-)-epicatechin (the present work, Fig. 3), and EGC, EGCG (see reference 15).
Point 4: Authors should mention the compound was synthesized or commercially purchased.
Response 4: See Materials and methods 4.1. Animals, mode of introduction of catechins and gallate, and aorta preparation, page 6: Matrix solutions (1 mg/ml) of all catechins (Sigma, USA)… and a matrix solution of gallate Na (Sigma, USA). This means that the compounds were commercially purchased.
Point 5: Authors guidelines should be followed references are also not in the format.
Response 5: Thank you. The format of references was corrected.
Point 6: Authors should give some possibe mechanism for the active catechin.
Response 6: A possible mechanism for the effect of catechin was suggested in Discussion (Page 5): “The (+)-catechin-induced increase in ACE activity in the aorta is caused by the adhesion of monocytes to the endothelium because the adhesion blocker fucoidin suppresses this increase.” We have no data for a more detail of this mechanism.
We are grateful to the reviewe for the critical remarks. Taking this helpful criticism into account substantially improved the manuscript.
The authors
Response to Reviewer 1 Comments
Point 1: Authors need to reduce the similiarity index which is more than 50%.
Response 1: The high similarity index is determined by the fact that materials and methods in this work are the same as in previous works. The section Materials and methods in this manuscript makes up 45% of the whole manuscript text. We cannot describe Materials and methods with other words and using another sequence of manipulations because we used the same animals, the same equipment, same drugs, and the same sequence of manipulations. Therefore, we deleted the detailed description of the methods and for the acquaintance with the methods and gave the references to previous works. The text in Introduction and Discussion marked in yellow was rewritten to reduce similarity. The similarity index also decreased as a consequence of an increased volume of the Introduction and Discussion sections.
Point 2: In the introduction authors should include the different bilogical activities posses by catechin authors may get benefitted by https://doi.org/10.3390/molecules24010156;https://doi.org/10.1002/bkcs.10108 ; https://doi.org/10.5012/jkcs.2021.65.2.106; etc.
Response 2: We included several reviews for the description of different biological activities of catechins and reference 1 - (-)-Epigallocatechin-3-Gallate Decreases Osteoclastogenesis …; reference 2 is not opened (error), and reference 3 is not related to the effects of natural catechins on biological processes. The text that was added in the manuscript: Population studies show that three cups of green or black tea a day significantly reduce the risk of CVDs [10]. Green tea contains about 30% of catechins of the dry weight of a leaf, and 75% of catechins are due to galloylated catechins: (–)-epigallocatechin galate and (–)-epigallocatechin [11]. Catechins possess various biological activities: anti-oxidative, antimicrobial, anti-allergic, anti-diabetic, anti-inflammatory, anti-cancer, chemoprotective, neuroprotective, and immunomodulatory [12, 13]. Epigallocatechin gallate reduces blood pressure, the level of blood cholesterol, the amount of body fat, and the risk of osteoporotic fractures [13, 14].
Point 3: Does dose dependent studies done for other compounds viz (-)- catechin etc
Response 3: Dose dependent studies were done for (+)-catechin, (-)-epicatechin (the present work, Fig. 3), and EGC, EGCG (see reference 15).
Point 4: Authors should mention the compound was synthesized or commercially purchased.
Response 4: See Materials and methods 4.1. Animals, mode of introduction of catechins and gallate, and aorta preparation, page 6: Matrix solutions (1 mg/ml) of all catechins (Sigma, USA)… and a matrix solution of gallate Na (Sigma, USA). This means that the compounds were commercially purchased.
Point 5: Authors guidelines should be followed references are also not in the format.
Response 5: Thank you. The format of references was corrected.
Point 6: Authors should give some possibe mechanism for the active catechin.
Response 6: A possible mechanism for the effect of catechin was suggested in Discussion (Page 5): “The (+)-catechin-induced increase in ACE activity in the aorta is caused by the adhesion of monocytes to the endothelium because the adhesion blocker fucoidin suppresses this increase.” We have no data for a more detail of this mechanism.
We are grateful to the reviewe for the critical remarks. Taking this helpful criticism into account substantially improved the manuscript.
The authors
Response to Reviewer 1 Comments
Point 1: Authors need to reduce the similiarity index which is more than 50%.
Response 1: The high similarity index is determined by the fact that materials and methods in this work are the same as in previous works. The section Materials and methods in this manuscript makes up 45% of the whole manuscript text. We cannot describe Materials and methods with other words and using another sequence of manipulations because we used the same animals, the same equipment, same drugs, and the same sequence of manipulations. Therefore, we deleted the detailed description of the methods and for the acquaintance with the methods and gave the references to previous works. The text in Introduction and Discussion marked in yellow was rewritten to reduce similarity. The similarity index also decreased as a consequence of an increased volume of the Introduction and Discussion sections.
Point 2: In the introduction authors should include the different bilogical activities posses by catechin authors may get benefitted by https://doi.org/10.3390/molecules24010156;https://doi.org/10.1002/bkcs.10108 ; https://doi.org/10.5012/jkcs.2021.65.2.106; etc.
Response 2: We included several reviews for the description of different biological activities of catechins and reference 1 - (-)-Epigallocatechin-3-Gallate Decreases Osteoclastogenesis …; reference 2 is not opened (error), and reference 3 is not related to the effects of natural catechins on biological processes. The text that was added in the manuscript: Population studies show that three cups of green or black tea a day significantly reduce the risk of CVDs [10]. Green tea contains about 30% of catechins of the dry weight of a leaf, and 75% of catechins are due to galloylated catechins: (–)-epigallocatechin galate and (–)-epigallocatechin [11]. Catechins possess various biological activities: anti-oxidative, antimicrobial, anti-allergic, anti-diabetic, anti-inflammatory, anti-cancer, chemoprotective, neuroprotective, and immunomodulatory [12, 13]. Epigallocatechin gallate reduces blood pressure, the level of blood cholesterol, the amount of body fat, and the risk of osteoporotic fractures [13, 14].
Point 3: Does dose dependent studies done for other compounds viz (-)- catechin etc
Response 3: Dose dependent studies were done for (+)-catechin, (-)-epicatechin (the present work, Fig. 3), and EGC, EGCG (see reference 15).
Point 4: Authors should mention the compound was synthesized or commercially purchased.
Response 4: See Materials and methods 4.1. Animals, mode of introduction of catechins and gallate, and aorta preparation, page 6: Matrix solutions (1 mg/ml) of all catechins (Sigma, USA)… and a matrix solution of gallate Na (Sigma, USA). This means that the compounds were commercially purchased.
Point 5: Authors guidelines should be followed references are also not in the format.
Response 5: Thank you. The format of references was corrected.
Point 6: Authors should give some possibe mechanism for the active catechin.
Response 6: A possible mechanism for the effect of catechin was suggested in Discussion (Page 5): “The (+)-catechin-induced increase in ACE activity in the aorta is caused by the adhesion of monocytes to the endothelium because the adhesion blocker fucoidin suppresses this increase.” We have no data for a more detail of this mechanism.
We are grateful to the reviewe for the critical remarks. Taking this helpful criticism into account substantially improved the manuscript.
The authors
Response to Reviewer 1 Comments
Point 1: Authors need to reduce the similiarity index which is more than 50%.
Response 1: The high similarity index is determined by the fact that materials and methods in this work are the same as in previous works. The section Materials and methods in this manuscript makes up 45% of the whole manuscript text. We cannot describe Materials and methods with other words and using another sequence of manipulations because we used the same animals, the same equipment, same drugs, and the same sequence of manipulations. Therefore, we deleted the detailed description of the methods and for the acquaintance with the methods and gave the references to previous works. The text in Introduction and Discussion marked in yellow was rewritten to reduce similarity. The similarity index also decreased as a consequence of an increased volume of the Introduction and Discussion sections.
Point 2: In the introduction authors should include the different bilogical activities posses by catechin authors may get benefitted by https://doi.org/10.3390/molecules24010156;https://doi.org/10.1002/bkcs.10108 ; https://doi.org/10.5012/jkcs.2021.65.2.106; etc.
Response 2: We included several reviews for the description of different biological activities of catechins and reference 1 - (-)-Epigallocatechin-3-Gallate Decreases Osteoclastogenesis …; reference 2 is not opened (error), and reference 3 is not related to the effects of natural catechins on biological processes. The text that was added in the manuscript: Population studies show that three cups of green or black tea a day significantly reduce the risk of CVDs [10]. Green tea contains about 30% of catechins of the dry weight of a leaf, and 75% of catechins are due to galloylated catechins: (–)-epigallocatechin galate and (–)-epigallocatechin [11]. Catechins possess various biological activities: anti-oxidative, antimicrobial, anti-allergic, anti-diabetic, anti-inflammatory, anti-cancer, chemoprotective, neuroprotective, and immunomodulatory [12, 13]. Epigallocatechin gallate reduces blood pressure, the level of blood cholesterol, the amount of body fat, and the risk of osteoporotic fractures [13, 14].
Point 3: Does dose dependent studies done for other compounds viz (-)- catechin etc
Response 3: Dose dependent studies were done for (+)-catechin, (-)-epicatechin (the present work, Fig. 3), and EGC, EGCG (see reference 15).
Point 4: Authors should mention the compound was synthesized or commercially purchased.
Response 4: See Materials and methods 4.1. Animals, mode of introduction of catechins and gallate, and aorta preparation, page 6: Matrix solutions (1 mg/ml) of all catechins (Sigma, USA)… and a matrix solution of gallate Na (Sigma, USA). This means that the compounds were commercially purchased.
Point 5: Authors guidelines should be followed references are also not in the format.
Response 5: Thank you. The format of references was corrected.
Point 6: Authors should give some possibe mechanism for the active catechin.
Response 6: A possible mechanism for the effect of catechin was suggested in Discussion (Page 5): “The (+)-catechin-induced increase in ACE activity in the aorta is caused by the adhesion of monocytes to the endothelium because the adhesion blocker fucoidin suppresses this increase.” We have no data for a more detail of this mechanism.
We are grateful to the reviewe for the critical remarks. Taking this helpful criticism into account substantially improved the manuscript.
The authors
Response to Reviewer 1 Comments
Point 1: Authors need to reduce the similiarity index which is more than 50%.
Response 1: The high similarity index is determined by the fact that materials and methods in this work are the same as in previous works. The section Materials and methods in this manuscript makes up 45% of the whole manuscript text. We cannot describe Materials and methods with other words and using another sequence of manipulations because we used the same animals, the same equipment, same drugs, and the same sequence of manipulations. Therefore, we deleted the detailed description of the methods and for the acquaintance with the methods and gave the references to previous works. The text in Introduction and Discussion marked in yellow was rewritten to reduce similarity. The similarity index also decreased as a consequence of an increased volume of the Introduction and Discussion sections.
Point 2: In the introduction authors should include the different bilogical activities posses by catechin authors may get benefitted by https://doi.org/10.3390/molecules24010156;https://doi.org/10.1002/bkcs.10108 ; https://doi.org/10.5012/jkcs.2021.65.2.106; etc.
Response 2: We included several reviews for the description of different biological activities of catechins and reference 1 - (-)-Epigallocatechin-3-Gallate Decreases Osteoclastogenesis …; reference 2 is not opened (error), and reference 3 is not related to the effects of natural catechins on biological processes. The text that was added in the manuscript: Population studies show that three cups of green or black tea a day significantly reduce the risk of CVDs [10]. Green tea contains about 30% of catechins of the dry weight of a leaf, and 75% of catechins are due to galloylated catechins: (–)-epigallocatechin galate and (–)-epigallocatechin [11]. Catechins possess various biological activities: anti-oxidative, antimicrobial, anti-allergic, anti-diabetic, anti-inflammatory, anti-cancer, chemoprotective, neuroprotective, and immunomodulatory [12, 13]. Epigallocatechin gallate reduces blood pressure, the level of blood cholesterol, the amount of body fat, and the risk of osteoporotic fractures [13, 14].
Point 3: Does dose dependent studies done for other compounds viz (-)- catechin etc
Response 3: Dose dependent studies were done for (+)-catechin, (-)-epicatechin (the present work, Fig. 3), and EGC, EGCG (see reference 15).
Point 4: Authors should mention the compound was synthesized or commercially purchased.
Response 4: See Materials and methods 4.1. Animals, mode of introduction of catechins and gallate, and aorta preparation, page 6: Matrix solutions (1 mg/ml) of all catechins (Sigma, USA)… and a matrix solution of gallate Na (Sigma, USA). This means that the compounds were commercially purchased.
Point 5: Authors guidelines should be followed references are also not in the format.
Response 5: Thank you. The format of references was corrected.
Point 6: Authors should give some possibe mechanism for the active catechin.
Response 6: A possible mechanism for the effect of catechin was suggested in Discussion (Page 5): “The (+)-catechin-induced increase in ACE activity in the aorta is caused by the adhesion of monocytes to the endothelium because the adhesion blocker fucoidin suppresses this increase.” We have no data for a more detail of this mechanism.
We are grateful to the reviewe for the critical remarks. Taking this helpful criticism into account substantially improved the manuscript.
The authors
Response to Reviewer 1 Comments
Point 1: Authors need to reduce the similiarity index which is more than 50%.
Response 1: The high similarity index is determined by the fact that materials and methods in this work are the same as in previous works. The section Materials and methods in this manuscript makes up 45% of the whole manuscript text. We cannot describe Materials and methods with other words and using another sequence of manipulations because we used the same animals, the same equipment, same drugs, and the same sequence of manipulations. Therefore, we deleted the detailed description of the methods and for the acquaintance with the methods and gave the references to previous works. The text in Introduction and Discussion marked in yellow was rewritten to reduce similarity. The similarity index also decreased as a consequence of an increased volume of the Introduction and Discussion sections.
Point 2: In the introduction authors should include the different bilogical activities posses by catechin authors may get benefitted by https://doi.org/10.3390/molecules24010156;https://doi.org/10.1002/bkcs.10108 ; https://doi.org/10.5012/jkcs.2021.65.2.106; etc.
Response 2: We included several reviews for the description of different biological activities of catechins and reference 1 - (-)-Epigallocatechin-3-Gallate Decreases Osteoclastogenesis …; reference 2 is not opened (error), and reference 3 is not related to the effects of natural catechins on biological processes. The text that was added in the manuscript: Population studies show that three cups of green or black tea a day significantly reduce the risk of CVDs [10]. Green tea contains about 30% of catechins of the dry weight of a leaf, and 75% of catechins are due to galloylated catechins: (–)-epigallocatechin galate and (–)-epigallocatechin [11]. Catechins possess various biological activities: anti-oxidative, antimicrobial, anti-allergic, anti-diabetic, anti-inflammatory, anti-cancer, chemoprotective, neuroprotective, and immunomodulatory [12, 13]. Epigallocatechin gallate reduces blood pressure, the level of blood cholesterol, the amount of body fat, and the risk of osteoporotic fractures [13, 14].
Point 3: Does dose dependent studies done for other compounds viz (-)- catechin etc
Response 3: Dose dependent studies were done for (+)-catechin, (-)-epicatechin (the present work, Fig. 3), and EGC, EGCG (see reference 15).
Point 4: Authors should mention the compound was synthesized or commercially purchased.
Response 4: See Materials and methods 4.1. Animals, mode of introduction of catechins and gallate, and aorta preparation, page 6: Matrix solutions (1 mg/ml) of all catechins (Sigma, USA)… and a matrix solution of gallate Na (Sigma, USA). This means that the compounds were commercially purchased.
Point 5: Authors guidelines should be followed references are also not in the format.
Response 5: Thank you. The format of references was corrected.
Point 6: Authors should give some possibe mechanism for the active catechin.
Response 6: A possible mechanism for the effect of catechin was suggested in Discussion (Page 5): “The (+)-catechin-induced increase in ACE activity in the aorta is caused by the adhesion of monocytes to the endothelium because the adhesion blocker fucoidin suppresses this increase.” We have no data for a more detail of this mechanism.
We are grateful to the reviewe for the critical remarks. Taking this helpful criticism into account substantially improved the manuscript.
The authors
Response to Reviewer 1 Comments
Point 1: Authors need to reduce the similiarity index which is more than 50%.
Response 1: The high similarity index is determined by the fact that materials and methods in this work are the same as in previous works. The section Materials and methods in this manuscript makes up 45% of the whole manuscript text. We cannot describe Materials and methods with other words and using another sequence of manipulations because we used the same animals, the same equipment, same drugs, and the same sequence of manipulations. Therefore, we deleted the detailed description of the methods and for the acquaintance with the methods and gave the references to previous works. The text in Introduction and Discussion marked in yellow was rewritten to reduce similarity. The similarity index also decreased as a consequence of an increased volume of the Introduction and Discussion sections.
Point 2: In the introduction authors should include the different bilogical activities posses by catechin authors may get benefitted by https://doi.org/10.3390/molecules24010156;https://doi.org/10.1002/bkcs.10108 ; https://doi.org/10.5012/jkcs.2021.65.2.106; etc.
Response 2: We included several reviews for the description of different biological activities of catechins and reference 1 - (-)-Epigallocatechin-3-Gallate Decreases Osteoclastogenesis …; reference 2 is not opened (error), and reference 3 is not related to the effects of natural catechins on biological processes. The text that was added in the manuscript: Population studies show that three cups of green or black tea a day significantly reduce the risk of CVDs [10]. Green tea contains about 30% of catechins of the dry weight of a leaf, and 75% of catechins are due to galloylated catechins: (–)-epigallocatechin galate and (–)-epigallocatechin [11]. Catechins possess various biological activities: anti-oxidative, antimicrobial, anti-allergic, anti-diabetic, anti-inflammatory, anti-cancer, chemoprotective, neuroprotective, and immunomodulatory [12, 13]. Epigallocatechin gallate reduces blood pressure, the level of blood cholesterol, the amount of body fat, and the risk of osteoporotic fractures [13, 14].
Point 3: Does dose dependent studies done for other compounds viz (-)- catechin etc
Response 3: Dose dependent studies were done for (+)-catechin, (-)-epicatechin (the present work, Fig. 3), and EGC, EGCG (see reference 15).
Point 4: Authors should mention the compound was synthesized or commercially purchased.
Response 4: See Materials and methods 4.1. Animals, mode of introduction of catechins and gallate, and aorta preparation, page 6: Matrix solutions (1 mg/ml) of all catechins (Sigma, USA)… and a matrix solution of gallate Na (Sigma, USA). This means that the compounds were commercially purchased.
Point 5: Authors guidelines should be followed references are also not in the format.
Response 5: Thank you. The format of references was corrected.
Point 6: Authors should give some possibe mechanism for the active catechin.
Response 6: A possible mechanism for the effect of catechin was suggested in Discussion (Page 5): “The (+)-catechin-induced increase in ACE activity in the aorta is caused by the adhesion of monocytes to the endothelium because the adhesion blocker fucoidin suppresses this increase.” We have no data for a more detail of this mechanism.
We are grateful to the reviewe for the critical remarks. Taking this helpful criticism into account substantially improved the manuscript.
The authors

This manuscript is a resubmission of an earlier submission. The following is a list of the peer review reports and author responses from that submission.